# Energy Efficiency for 5G and Beyond 5G: Potential, Limitations, and Future Directions

**DOI:** 10.3390/s24227402

**Published:** 2024-11-20

**Authors:** Adrian Ichimescu, Nirvana Popescu, Eduard C. Popovici, Antonela Toma

**Affiliations:** Faculty of Automatic Control and Computer Science, Computer Science Department, National University of Science and Technology POLITEHNICA Bucharest, 060042 Bucharest, Romania; adrian.ichimescu@ubp.ro (A.I.); eduard.popovici@upb.ro (E.C.P.); antonela.toma@upb.ro (A.T.)

**Keywords:** 5G, energy efficiency, sleep modes, renewable energy, traffic offloading, clustering

## Abstract

Energy efficiency constitutes a pivotal performance indicator for 5G New Radio (NR) networks and beyond, and achieving optimal efficiency necessitates the meticulous consideration of trade-offs against other performance parameters, including latency, throughput, connection densities, and reliability. Energy efficiency assumes it is of paramount importance for both User Equipment (UE) to achieve battery prologue and base stations to achieve savings in power and operation cost. This paper presents an exhaustive review of power-saving research conducted for 5G and beyond 5G networks in recent years, elucidating the advantages, disadvantages, and key characteristics of each technique. Reinforcement learning, heuristic algorithms, genetic algorithms, Markov Decision Processes, and the hybridization of various standard algorithms inherent to 5G and 5G NR represent a subset of the available solutions that shall undergo scrutiny. In the final chapters, this work identifies key limitations, namely, computational expense, deployment complexity, and scalability constraints, and proposes a future research direction by theoretically exploring online learning, the clustering of the network base station, and hard HO to lower the consumption of networks like 2G or 4G. In lowering carbon emissions and lowering OPEX, these three additional features could help mobile network operators achieve their targets.

## 1. Introduction

According to the GSMA [1], the telecom industry is responsible for 2–3% of global energy consumption, and power costs constitute 15–40% of an operator’s operating expenses (OPEX). Consequently, major telecom providers like Deutsche Telekom, Vodafone, Verizon, AT&T, and Orange have established ambitious targets for reducing power consumption and carbon emissions in the near future [2,3,4,5,6].

Networks consume 90% of a telecom operator’s total power/electricity, with 60–80% attributed to the Radio Access Network (RAN), as per GSMA data [7]. The 3GPP [8] defines network energy efficiency as the amount of data transmitted per unit of energy consumed, measured in bits per Joule (bit/J). A higher bit/J value signifies greater energy efficiency.

5G RAN, depicted in Figure 1, has substantial potential for energy savings and has become a focal point for research. Various approaches have been proposed, including the following:Integrating renewable energy sources with traffic offloading and Advanced Sleep Modes [9];Implementing mechanisms to deactivate underutilized small cells based on predicted user mobility [10,11];Utilizing machine learning to control cell sleep modes efficiently based on traffic, interference, and buffer status [12,13];Developing heuristic solutions that consider energy efficiency and power allocation by switching off base stations [14];Proposing algorithms to dynamically adjust cell coverage and base station activity based on traffic load, maximize energy efficiency, and balance grid and solar power consumption [15];Employing genetic algorithms to deactivate underutilized base stations [16];Investigating dynamic switching between base station backhauling technologies [17];Optimizing Advanced Sleep Mode (ASM) parameters to balance energy savings and delay constraints, addressing a new aspect of Quality of Service (QoS) [18];Proposing a clustering-based energy-saving scheme to minimize active base stations while ensuring QoS for all User Equipment (UE) [19];Developing a centralized dynamic sleep method using a genetic algorithm and clustering to enhance energy savings through coordinated sleep decisions [20];Dynamically updating the sleep mode status of Radio Units (RUs) based on the network state to ensure that only necessary RUs remain active, meeting demand and maintaining QoS [21];Dynamically controlling power-saving modes using graph theory to determine the order of cell activation/deactivation, maximizing power savings while maintaining coverage and minimizing signaling [22].

The need for user connectivity and communication capabilities in any geographical area where 5G or traditional mobile technologies cannot be made available is continuously rising. That is the reason why advanced network architectures like Satellite–Terrestrial Integrated Networks (STINs), depicted in Figure 2, and Integrated Terrestrial–Aerial Networks (ITANs), depicted in Figure 3, have begun to attract more and more interest. The key difference between STINs and ITANs lies in ITANs’ integration of the aerial platform and utilization of RISs (reconfigurable intelligent surfaces), while STINs are provided by terrestrial and satellite networks with a common scope of delivering ubiquitous communication improving security and energy consumption.

Various approaches to achieve energy efficiency for beyond 5G new networks have been taken:Hybrid beamforming design in [23,24];Robust beamforming design in [25].

Besides these, other approaches look at overall energy efficiency solutions by focusing on computational load, indirectly obtaining 5G networks’ energy efficiency [26,27].

Section 2 of this paper will provide a detailed analysis of these solutions, establishing criteria for comparison and identifying their strengths and weaknesses, which will be discussed in Section 3. Section 4 will conclude with potential enhancements.

## 2. Energy Efficiency Solutions Available in the Market

The rapid evolution of mobile networks, from 4G to 5G and beyond, has brought about a tremendous surge in data traffic and connected devices. This growth, while enabling exciting new possibilities, also presents a critical challenge: energy consumption. As networks expand and densify to meet ever-increasing demand, their energy footprint grows significantly, raising concerns about environmental impact and operational costs. Therefore, a crucial focus for the future of mobile connectivity lies in developing and implementing innovative solutions to optimize energy efficiency across 5G and beyond networks like ITANs and STINs. This involves not only improving the energy performance of individual network components, but also rethinking network architectures, protocols, and deployment strategies to minimize overall energy usage without compromising performance or user experience.

In this context, the work of Adil Israr et al. [9] has addressed the escalating energy consumption in 5G networks, triggered by the surge in 5G and IoT devices. They propose a holistic solution centered around integrating renewable energy sources, intelligent traffic management, and advanced power-saving techniques. To reduce reliance on the traditional carbon-intensive grid, the authors advocate incorporating renewable energy sources like solar and wind power into the 5G infrastructure. This hybrid energy paradigm aims to decrease both operational costs and environmental impact. Furthermore, to optimize energy use during peak hours, the solution dynamically offloads user traffic from power-hungry macro base stations to energy-efficient small cell base stations.

During off-peak hours, underutilized base stations are intelligently put into varying levels of sleep modes, conserving energy by gradually deactivating components. A centralized renewable energy farm is also envisioned to provide a cost-effective and reliable power source for both macro and small cell base stations. The complex dynamics of energy supply, consumption, storage, and traffic patterns are analyzed using stochastic modelling, providing insights for optimizing energy-saving strategies. The proposed algorithm strikes a balance between energy efficiency and Quality of Service (QoS) by employing traffic-aware offloading, Advanced Sleep Modes, and Markov Chain Modelling to analyze the probabilities of, and transition rates between, different sleep states.

The algorithm’s objectives are to minimize blocking probability and reactivation delays while maximizing energy savings. The hybrid energy supply system ensures network availability and QoS even during periods of low renewable energy generation. The effectiveness of this solution is gauged using key metrics like energy gain, overall energy consumption, blocking probability, and reactivation delay.

Beyond direct energy savings, the framework explores the concept of *virtual energy cooperation*, where surplus renewable energy from one base station can be shared with others experiencing high traffic demand. This comprehensive approach not only tackles the pressing issue of 5G network energy consumption but also paves the way for a more sustainable and resilient future for mobile networks.

In the realm of energy-efficient 5G networks, Hasan Farooq et al. [10] introduced AURORA, a framework designed to proactively conserve energy in future ultra-dense 5G environments. AURORA’s core innovation lies in its ability to predict user mobility patterns and preemptively place small cells into sleep mode when they are anticipated to be underutilized.

By analyzing historical handover traces, AURORA employs a semi-Markov model to forecast not only the next cell a user is likely to move to but also the estimated handover time. Further enhancing its predictive capabilities, the framework incorporates “landmarks” to approximate future user locations. Based on these predicted user locations and cell loads, AURORA formulates an optimization problem to minimize energy consumption by strategically deactivating underutilized small cells. To ensure a balanced load across active cells and maintain Quality of Service (QoS), it also utilizes Cell Individual Offsets (CIOs).

Extensive simulations with realistic mobility traces have showcased AURORA’s ability to significantly reduce energy consumption without compromising QoS. It outperforms traditional reactive approaches and maintains energy savings even in the face of minor inaccuracies in mobility prediction. AURORA’s commitment to QoS is evident in its explicit focus on balancing energy efficiency with user experience. It achieves this by ensuring a fair distribution of load across active cells through load balancing with CIOs, which prevents overload and maintains service quality. Additionally, a Minimum Rate Constraint guarantees a minimum data rate for each user, ensuring sufficient resources even when some cells are in sleep mode. A Coverage Constraint is also maintained to ensure reliable network access across the coverage area, even with some cells inactive. The Load Threshold parameter provides network operators with the flexibility to adjust the balance between energy saving and QoS based on their specific needs. Through these mechanisms, AURORA aims to strike an optimal balance between energy efficiency and user satisfaction, ensuring a sustainable and high-performing 5G network for the future.

Fateh Elsherif et al. [11] have proposed a novel energy-saving approach for 5G cellular networks that involves dynamically controlling the on/off status of base stations (BSs) in real time. The approach formulates the problem as a Markov Decision Process (MDP), where the system state encompasses user positions, velocities, and BS statuses. The algorithm selects actions (turning BSs on or off) to minimize total energy consumption, including operational and switching costs. The authors introduce a policy rollout algorithm with Monte Carlo sampling to solve the MDP and propose a modified action space for improved computational efficiency. Simulations demonstrate significant energy savings compared to benchmark schemes, showcasing the algorithm’s effectiveness in various scenarios.

While the study primarily focused on simulation-based evaluation, the authors addressed practical implementation challenges. A modified action space reduces computational complexity, and the policy rollout algorithm is inherently efficient. Simulation results further indicated adaptability to real-world network conditions. Although direct resource consumption analysis in real networks was absent, the emphasis on computational efficiency and adaptability suggested the possibility of practical deployment. Further research and real-world experiments are needed to quantify resource requirements and assess large-scale feasibility.

Related work using MDP for BS on/off switching exists, with energy efficiency evaluated against realistic traffic patterns in ns-3 simulations [28]. MDP’s ability to identify optimal decisions in dynamic systems makes it attractive for managing 5G networks with high user mobility. This is aligned with the conclusions drawn by Junhyuk Kim et al. [29] regarding MDP’s potential for achieving network energy efficiency.

Ali El Amine et al. [12] have proposed a reinforcement learning-based approach to optimize energy consumption in 5G Heterogeneous Networks (HetNets) by dynamically adjusting small base station (SBS) sleep modes. The core idea is to enable SBSs to switch between various sleep levels based on traffic load, interference, and buffer status, balancing energy efficiency with Quality of Service (QoS).

The problem is formulated as a Markov Decision Process (MDP), where each SBS acts as an agent learning the optimal sleep mode policy. The SBS’s state is defined by throughput, buffer size, and estimated interference. Available actions are different sleep mode levels, including active and deeper energy-saving modes. The proposed Q-learning algorithm allows SBSs to learn the optimal policy by exploring actions and observing their impacts. A weighting parameter balances energy saving and QoS priorities. Simulations demonstrated significant energy savings compared to baseline schemes. The algorithm adapts to varying traffic and interference, ensuring appropriate sleep modes for optimal balance. Investigating the impact of user offloading from inactive SBSs to the macro base station (MBS) revealed a potential trade-off between reduced data loss/delay and decreased overall energy efficiency due to higher MBS power consumption.

While primarily theoretical and simulation-based, the approach suggests real-world potential due to its distributed decision-making, limited action space, and consideration of practical factors. However, challenges like hardware requirements, coordination, learning time, and security/reliability need to be addressed for real-world deployment.

Further research is required to assess feasibility and effectiveness in real networks. Future work should focus on overcoming practical challenges and adapting the algorithm for specific real-world scenarios.

To tackle the challenge of energy consumption in 5G millimeter wave networks, Abdulhalim Fayad et al. [14] have leveraged the power of a Genetic Algorithm (GA). This heuristic approach seeks to minimize power consumption while upholding Quality of Service (QoS) by intelligently assigning users to base stations and allocating power resources.

The GA operates by first generating a diverse population of potential solutions. Each solution represents a specific user–base station assignment and power allocation scheme. These solutions are then evaluated using a fitness function that considers power consumption and QoS factors like active base stations, transmission power, and user data rates. The GA favors solutions with lower power consumption and acceptable QoS, selecting them for the next generation. New solutions are created through crossover (combining elements of selected solutions) and mutation (randomly modifying solutions), ensuring a broad exploration of the solution space. This iterative process continues refining the solutions over generations until the GA converges on the optimal or near-optimal user–base station assignment and power allocation scheme for energy efficiency.

Simulations have showcased the GA’s effectiveness in curbing 5G mmWave network power consumption, especially in scenarios with a large number of users where exact methods become computationally prohibitive. While the solution does not explicitly address user mobility, the inherent high density of base stations in mmWave networks may mitigate the impact of fast user movement. This density increases the likelihood of users remaining within a base station’s coverage or experiencing shorter handoffs, potentially enhancing the algorithm’s robustness to mobility. However, further research and real-world testing are necessary to definitively assess the algorithm’s performance under fast user mobility scenarios.

Abu Jahid et al. [15] have proposed a novel approach to improve energy efficiency in green cellular networks, focusing on cloud radio access networks (C-RANs) powered by renewable energy sources. Their solution involves a dynamic point selection coordinated multipoint (DPS CoMP)-based load-balancing scheme to optimize throughput and energy efficiency by minimizing reliance on the traditional power grid.

This approach addresses the intermittent nature of renewable energy by integrating it with the grid and tackles spatial/temporal traffic variations by dynamically adjusting cell coverage and base station activity based on real-time traffic demand. A heuristic load-balancing algorithm intelligently associates users with base stations based on signal quality and traffic load, enabling offloading or sleep modes for underutilized base stations. While not explicitly addressing real-world resource utilization, the paper acknowledges potential challenges like computational complexity, communication overhead, hardware requirements, scalability, real-time adaptability, and the energy overhead of coordination. Despite these challenges, the proposed heuristic algorithm aims for computational efficiency and real-time adaptability. Simulation results show promising energy savings. Further research and real-world trials are necessary to fully assess its feasibility and resource requirements in large-scale deployments.

Silvestre Malta et al. [13] have proposed using the SARSA reinforcement learning algorithm to manage 5G base station sleep modes, optimizing energy consumption while maintaining Quality of Service (QoS). The approach considers various 5G use cases and their latency requirements. The system model, designed to optimize energy efficiency in 5G networks, incorporates traffic modulation, energy consumption, and sleep mode policies. Traffic patterns were simulated using a Poisson process, while the energy model accounts for the time spent in each sleep mode and the energy expended during transitions. The SARSA (State–Action–Reward–State–Action) algorithm is employed to learn the optimal sleep mode actions in various states, balancing energy savings with latency penalties.

SARSA, an on-policy temporal difference learning algorithm, learns the optimal actions to take in different states to maximize rewards. States are defined by the current sleep mode level and packet buffer load, while actions correspond to the different sleep modes the base station (BS) can switch to. The algorithm assigns rewards for energy savings and penalties for delays or high buffer loads. A Q-table stores each state–action pair’s estimated values (Q-values), which represent the expected future rewards. The policy for selecting actions is then updated based on these Q-values, striking a balance between exploration (trying new actions) and exploitation (choosing actions known to yield high rewards).

Simulations demonstrated that significant energy savings are achievable, particularly in low-traffic scenarios or when prioritizing energy reduction. Savings of up to 80% are possible while still meeting latency requirements. The system’s adaptability to varying traffic and latency constraints enables energy optimization without sacrificing QoS.

The algorithm’s behavior adapts to the specific QoS requirements of different 5G use cases. In eMBB (enhanced Mobile Broadband) scenarios, where high data rates are prioritized, significant energy savings are possible, especially with relaxed latency thresholds. In mMTC (massive Machine-Type Communication) scenarios, involving many devices with low data rates and moderate latency tolerance, energy savings are achievable but may be lower than in eMBB. URLLC (Ultra-Reliable Low-Latency Communication), with its stringent latency and reliability demands, allows for energy savings, but these are generally lower, especially with strict latency thresholds.

The system’s adaptability to latency thresholds allows it to cater to the unique requirements of each use case. The “Buflim” parameter, which controls the maximum buffer latency, provides further flexibility in prioritizing either energy savings or QoS. While the focus of this work is on theoretical modeling and simulation, the authors suggest potential real-world applicability due to the algorithm’s adaptability to varying traffic and latency constraints. However, practical challenges such as hardware requirements, coordination among base stations, learning time for the algorithm, and ensuring security and reliability need further exploration before real-world deployment.

In their work, Hasna Fourati et al. [16] presented ESGA-5G, an energy-saving scheme that leverages the power of a Genetic Algorithm (GA) to optimize the on/off states of small base stations (SBSs) in 5G Heterogeneous Networks (HetNets). The central objective was to strike a balance between minimizing energy consumption and satisfying user traffic demand. ESGA-5G’s implementation starts with the creation of a random population of potential solutions, where each solution, represented as a chromosome, encodes the on/off states of the SBSs. These solutions are evaluated using a fitness function that takes into account both energy consumption and penalties for any unmet traffic demand. Chromosomes with higher fitness values, signifying lower energy consumption and fulfilled traffic demand, are selected as parents for the next generation using tournament selection. The genetic information of these parent chromosomes is then combined through crossover to create offspring. To introduce diversity and prevent the algorithm from becoming stuck in local optima, random bit flips, or mutations, are applied to the offspring.

The resulting offspring, along with some of the top-performing parents, form the new population, replacing the old one. This iterative process of evaluation, selection, crossover, and mutation continues until a satisfactory solution is found or a predetermined number of generations is reached. The best solution emerging from this process represents the near-optimal on/off states for the SBSs, achieving minimal energy consumption while fulfilling traffic demand.

Results showed that ESGA-5G outperforms Particle Swarm Optimization (PSO) in terms of energy savings, underscoring the potential of Genetic Algorithms for energy optimization in the complex landscape of 5G HetNets.

In their quest for greener 5G networks, Munjure Mowla and a team of researchers [17] embarked on a journey to optimize energy efficiency in the intricate world of small cell networks (SCNs). They envisioned a hybrid backhauling approach, skillfully weaving together the strengths of passive optical networks (PONs) and millimeter wave (mmWave) technologies. A PON, with its shared medium and passive components, shines under the pressure of heavy traffic, while mmWave, offering high data rates and flexibility, proves more energy-conscious during periods of low load.

The team’s challenge was to create an adaptive strategy that could seamlessly switch between PON and mmWave depending on the ever-changing traffic patterns, ensuring that power consumption remained minimal while data rates were consistently met. This led them to formulate an optimization problem, seeking the most energy-efficient path for each traffic scenario. However, the complexity of this problem demanded a more practical solution.

Thus, a heuristic approach was born. By utilizing a simplified energy consumption model, they developed a computationally efficient method to determine the optimal switching threshold in real time. This allowed the network to make intelligent decisions on the fly, always selecting the most energy-efficient backhauling technology for the current traffic conditions.

The results were impressive. Compared to traditional single-technology approaches, the hybrid approach showcased significant energy savings, reaching up to 32% in some scenarios. It offered a level of adaptability and flexibility that was previously unattainable, proving that the future of 5G networks could be both powerful and sustainable.

In their quest for greener 5G networks, Daniela Renga et al. in [18] unveiled DCASM, a clever strategy to conserve energy in 5G base stations without sacrificing performance. DCASM harnesses the power of Advanced Sleep Modes (ASMs), allowing base stations to slip into deeper states of slumber when traffic is light, thus slashing energy consumption. However, with deeper sleep comes a longer wake-up call, a trade-off that DCASM deftly manages. At the heart of DCASM lies a delicate balancing act between energy savings and delay constraints. The system taps into real-world mobile traffic patterns and employs an ANN-based prediction algorithm to anticipate future demand. Based on the application’s sensitivity to delays, it establishes a maximum average time for a base station to rouse itself from its sleep. The crux of the matter is optimizing the “hold time” (T2)—the duration spent in the intermediate sleep mode, sleep mode 2 (SM2), before descending into the deepest sleep mode, sleep mode 3 (SM3).

DCASM’s ingenuity lies in its dynamic adaptability. During periods of low traffic, it encourages base stations to sleep longer, maximizing energy conservation. Conversely, as traffic surges, it ensures they remain in lighter sleep states or even fully awake, ready to respond promptly. This dynamic approach guarantees that average wake-up times stay within the predefined limits, making it ideal even for applications that demand swift responses. The brilliance of DCASM extends beyond theory. A closed-form expression enables the calculation of optimal sleep durations, streamlining its implementation in real 5G base stations. This practicality, combined with its ability to significantly reduce energy consumption while upholding performance guarantees, positions DCASM as a promising solution for a sustainable 5G future.

In the intricate landscape of 5G dense small cell networks (SCNs), Wei Kuang Lai et al. [19] envisioned a power-saving strategy that revolves around the concept of clustering. The goal is ambitious: to selectively switch off base stations, thus minimizing energy consumption, while ensuring that users continue to enjoy a satisfactory Quality of Service (QoS). Recognizing the computational challenge of finding the optimal on/off configuration for base stations (BSs), Lai proposes a three-phase heuristic approach. The first phase involves partitioning the network into clusters based on BS load. Heavily burdened BSs are designated as the cores of these clusters. Next, within each cluster, an exhaustive search is conducted to identify the ideal combination of active and inactive BSs. This search aims to minimize the number of active BSs while still upholding the required QoS. Finally, the algorithm extends its reach to the boundaries between clusters, evaluating whether users served by BSs on the edges could be handed over to neighboring clusters, further reducing energy consumption.

The clustering algorithm itself is a well-orchestrated process. It begins by calculating the load on each BS, taking into account the resource blocks (RBs) demanded by its users and the total available RBs. BSs are then classified as underloaded, medium loaded, or overloaded using predefined thresholds. Heavily loaded BSs, or the two most loaded ones if none meet the “heavily loaded” criteria, are chosen as cluster cores. The remaining BSs then align themselves with the nearest core BS, forming clusters. Should any cluster become too large, it is split into two, with the two most loaded BSs within it assuming the role of new cores.

Through simulations, this clustering-based scheme has proven its ability to significantly reduce power consumption without compromising QoS. Its performance shined particularly in scenarios where users were not uniformly distributed, outperforming existing approaches. In essence, this heuristic clustering algorithm serves as a powerful tool, effectively dividing a network into manageable subproblems, enabling the efficient on/off control of BSs within each cluster, and paving the way for a more energy-conscious 5G future.

Mosheer J. Daas et al. [22] have proposed a novel energy management framework for 5G ultra-dense networks (UDNs) using graph theory. The framework is designed to address the increased energy demands of 5G networks due to the integration of small cells (SCs) alongside macro cells (MCs). The key idea is to dynamically control power-saving modes in the radio network by modeling the network as a graph and then using graph theory methods to determine the order in which nodes (cells) are switched off or on.

The algorithm, called STAR5, prioritizes nodes for power-saving based on their traffic load and type (SC or MC). It aims to maximize power savings while maintaining full network coverage and minimizing control plane signaling. The paper evaluated the algorithm’s performance under different network densification levels, load factors, and real-life network scenarios. The results showed that the proposed framework could achieve significant power savings, up to 25% at full load and 65% during off-peak hours, without compromising network performance.

In this groundbreaking work, the authors embarked on a journey to model complex 5G ultra-dense networks (UDNs) using the elegant language of graph theory. They constructed a sophisticated energy management framework upon this foundation, paving the way for intelligent power control. The power of graph theory was further harnessed to determine the sequence in which nodes should be powered off or on, employing the Weighted Degree Centrality metric as a guiding principle. Through rigorous evaluation under diverse network conditions, the authors showcased the efficacy of their algorithm in achieving substantial power savings, underscoring its potential to revolutionize energy efficiency in the realm of 5G UDNs.

Overall, the paper presents a promising approach to energy management in 5G UDNs, leveraging graph theory to optimize power consumption while maintaining network performance.

Kuo-Chi Chang et al. [20] have proposed an energy-saving technology for 5G base stations using Internet of Things (IoT) collaborative control. It addresses the issue of high energy consumption in dense 5G networks, particularly during periods of low traffic. The technology involves dynamically putting low-load base stations into a sleep mode to conserve energy.

The paper introduced a centralized dynamic sleep method based on a genetic algorithm. This method considers all possible combinations of base station sleep states to find the optimal configuration that minimizes energy consumption while maintaining network performance. The genetic algorithm is used to efficiently search for the best solution in a large solution space.

The proposed approach also includes a clustering algorithm that groups base stations based on their cooperation factors. This allows for coordinated sleep decisions within clusters, further enhancing energy savings. The clustering algorithm is dynamic, adapting to changes in network traffic and user demand.

Simulation results demonstrated the effectiveness of the proposed technology in reducing energy consumption and improving energy efficiency in 5G base station networks. The centralized sleep strategy based on the genetic algorithm is shown to be effective in finding optimal sleep combinations, while the clustering algorithm helps to reduce computational complexity, making the approach more practical for real-world deployment.

Kooshki et al. [13] introduced the 3xE scheme, an energy-efficient approach for cell-less radio access networks (RANs) in 5G and beyond. This scheme aims to optimize energy use by selectively deactivating underutilized access points (APs) while maintaining Quality of Service (QoS).

The algorithm considers both interference and load when deciding which APs to put into sleep mode. It first identifies APs that contribute more interference than useful signal and deactivates them. Then, it evaluates the remaining APs based on their load and impact on overall energy efficiency, further optimizing the network’s energy usage.

This algorithm can be classified as a heuristic optimization algorithm, as it aims to find a good solution efficiently rather than guaranteeing the absolute optimal solution. It incorporates elements of greedy algorithms and iterative improvement to achieve this goal.

The algorithm also takes into account the QoS provided to customers by including a constraint that ensures each user’s throughput meets a minimum requirement. By optimizing energy usage and considering QoS, the algorithm aims to improve the overall network performance and user experience.

Zhi Lin et al. in [23] introduced two novel and robust beamforming (BF) schemes to improve the secrecy energy efficiency (SEE) of Satellite–Terrestrial Integrated Networks (STINs) in mmWave band shared with cellular systems. The SEE metric is evaluated to gauge performance and is defined as the ratio of the achievable sum rate to the total power consumption. The element of novelty of the proposal is achieved through the following: hybrid analog–digital beamforming, robust design for the imperfect knowledge of the AoDs (angles of departure) for wiretap channels, leveraging interference from BSs to enhance the SEE performance of ESs (earth stations), optimization algorithms such as the Charnes–Cooper approach and the SCA (sequential convex approximation) method to solve the nonconvex optimization problem associated with hybrid beamforming design, and joint optimization of the BF weight vectors at the satellite.

While the authors acknowledge the computational complexity, they highlight the importance of selecting feasible initial points for the iterative optimization algorithms. One key aspect of the solution is that it takes into consideration realistic scenarios involving both single and multiple ESs. For a single ES, the authors propose using an iterative optimization algorithm that involves two stages: first obtaining the digital beamformer using the Charnes–Cooper approach, with auxiliary variables converted into second-order cone (SOC) constrains solved using semi-definite relaxation (SDR) and randomization techniques, and, second, using bisection search and SDR to obtain the analog beamformer. To further improve the solution, the authors utilize the iterative penalty function approach to minimize the difference between the trace and maximum eigenvalue of the beamforming matrices, ensuring rank-1 constraint. For more complex scenarios with multiple ESs, the authors adopt the SCA method, approximating original non-convex problems with a series of convex problems. The algorithm iteratively solves the convex problems, refining the solution until convergence. Similarly to the case of a single ES, the authors optimize digital and analog beamformers by adding steps to handle multi-user scenarios, including auxiliary variables and first-order Taylor series expansions to approximate non-convex constraints.

This solution takes into consideration practical constraints like imperfect channel state information (CSI), the use of massive antenna arrays due SNIR requirements, and RF chain limitation at the base station. Simulation results showed that the proposed hybrid beamforming approach consistently outperforms digital beamforming, and interleaved and localized hybrid architecture has better performance compared to fully connected architecture.

A similar approach involving beamforming techniques, which also takes into consideration jamming and eavesdropping attacks, is provided by Yifu Sun et al. in [24]. They propose a block coordinate descent (BCD) framework that jointly optimizes the user’s received decoder, the terrestrial and aerial digital precoders, and the multilayer RIS analog precoder. The BCD framework includes techniques like the heuristic beamforming scheme, SCA approach, and a monotonic vertex-update algorithm with a penalty convex-concave procedure (P-CCP), to address the non-convexity of the optimization problem. This approach also takes into consideration the imperfection of CSI.

Simulation results have shown an enhancement of 0.0814 and 0.1190 bps/Hz/Joule in comparison with the SDR-based BF scheme and SCA-based BF scheme, respectively.

Although energy efficiency is not their main focus, Hehao Niu et al. in [25] indirectly contribute to improving it by considering the use of artificial noise (AN) and phase shifters. These techniques can reduce the power consumption of intelligent reflecting surface (IRS)-assisted simultaneous wireless information and power transfer SWIPT networks, maintaining a secrecy performance that is the prime objective. An IRS-assisted secrecy SWIPT network is a wireless communication system that uses an IRS to improve the security of SWIPT. The authors’ solution is focused on maximizing the minimum robust information rate among the legitimate information receivers (IRs) in the presence of imperfect CSI.

Badie et al. in [26] and Abdulwahib et al. in [27] obtained 5G network energy efficiency indirectly by saving the overall solution computational load in vehicular network authentication. The former accomplished this by focusing on reducing re-authentication overhead through the bilinear pairing of points on the elliptic curve and blockchain technology, while the latter employed a combination of lightweight cryptography, optimized design, and fog computing.

## 3. Identifying Pros and Cons of Available Solutions

The algorithms designed to enhance 5G and beyond 5G networks’ energy efficiency analyze several key features:Mobility prediction: anticipating user movement to optimize resource allocation;Traffic offloading: shifting users from congested base stations to less busy ones;Sleep modes: putting base stations into low-power states during periods of inactivity;Renewable energy: utilizing solar or wind power to supplement the network’s energy supply;Clustering: grouping base stations based on load or cooperation factors to simplify algorithm complexity;Hybridization of the solutions: using multiple algorithms, obtaining the value of their benefits while lowering limitations;Optimization of the solution computational load: reducing the number of operations in the solution transaction.

These features are integrated into various algorithms, often employing reinforcement learning, genetic algorithms, or heuristic solutions, to achieve not only energy efficiency but also to maintain Quality of Service (QoS) for users and the security of communication.

Table 1, Features of energy efficiency studies of 5G networks, summarizes the different algorithms, highlighting that most leverage sleep modes, a standard 5G power-saving mechanism. A few solutions also explore alternative approaches like adjusting backhaul technologies based on network load or simply switching base stations on/off.

Table 2, Features of studies of beyond 5G new networks, compares different approaches of new network architectures to improve energy efficiency through hybrid beamforming design while providing communication capabilities similar to cellular networks in any geographical area.

Further enhancements have been incorporated to improve QoS, reduce algorithm complexity, or further boost energy efficiency. These include the following:Renewable energy integration: supplementing power supply during peak load times with solar energy;Clustering: simplifying algorithms by grouping base stations;Traffic offloading: ensuring user QoS while improving efficiency by moving users from low-traffic to high-traffic base stations;Mobility prediction: predicting traffic load to optimize base station sleep modes.

Implementing any of these energy-efficient algorithms can lead to significant energy savings ranging from 10% to 80%. The actual savings depend on factors like peak hours, network density, algorithm parameters, coordination between base stations and the central controller, network topology, and latency.

In Table 3 and Table 4, we have analyzed the advantages and limitations of each studied solution, and motivated by our findings, we propose a hybrid solution of the studied ones in order to achieve better energy efficiency overall. In future works, we aim to measure the results of our solution against a data set of data records tracking resources available for cells of a 2G, 4G, and 5G real network. Similarly, we aim to benchmark our proposed solution against the studied ones in this article having the same data set.

Our main contributions are summarized below:Examining the literature and identifying available solutions in the last five years;Identifying the advantages and limitations of available solutions;Identifying a theoretical solution to improve the energy efficiency of communication networks through the hybridization of available solutions, highlighting unexplored areas that will be measured in future studies.

## 4. Possible Areas of Improvement

In forthcoming investigations, we aim to mitigate the identified limitations of existing 5G networks and beyond 5G new networks’ energy efficiency solutions, as outlined in Table 3, Advantages and limitations of energy efficiency studies of 5G networks, and Table 4, Studies’ comparisons of beyond 5G new networks. Specifically, we recognize the following challenges:Solutions based on reinforcement learning [12,13] necessitate a learning phase, potentially delaying implementation;Approaches like [11,14,16,22] may entail high computational complexity, posing scalability concerns;Heuristic algorithms such as [14,15,17,18,21] may not guarantee optimal solutions;Some solutions necessitate complex infrastructure, such as renewable power sources [9,15] or specific backhauling technologies [17];Approaches need hybrid solutions to achieve maximum improvement of EE [24,25];Approaches need to take into consideration real network constraints and complexities like the imperfection of CSI, different ES scenarios [24,25], or computational load [26,27].

Our future research will explore strategies to overcome these constraints or use the advantages and develop more efficient, scalable, and readily deployable energy-saving mechanisms for cellular networks.

The inherent complexity of networks, coupled with high user mobility, data throughput demands, and base station density, pose challenges in implementing algorithms that efficiently manage these factors. Existing approaches may lead to computationally intensive and potentially suboptimal solutions.

This study has helped us establish the next step in our future research and has hopefully helped other researchers obtain a clear overview of the discussed topic. Thus, our goal will be to investigate online algorithms to determine optimal base station sleep mode strategies. We will evaluate the impact of these algorithms on energy efficiency, network signaling overhead, and user Quality of Service (QoS). Furthermore, we plan to explore the hybridization of online algorithms with other techniques such as clustering the BSs and HOs of users to lower consumption networks and renewable energy utilization, if available.

To ensure practical relevance, we aim to measure the energy efficiency of these strategies and the algorithm’s time to convergence, and to compute power needs using real-world network data collected with a service assurance solution monitoring a real 2G, 4G, and 5G core network. This data-driven approach will allow us to validate our findings and provide insights into the efficacy of proposed solutions in real-world scenarios.

To make the theoretical solution more clear, please refer to the block diagram depicted in Figure 4.

## 5. Conclusions

The exploration of energy efficiency solutions for 5G and beyond 5G networks has revealed a diverse range of innovative approaches, each with its own strengths and limitations. The surveyed research underscores the concerted effort to address these networks’ escalating energy demands, driven by the surge in data traffic, device proliferation, and the integration of heterogeneous technologies. The proposed solutions leverage a spectrum of techniques, including sleep modes, traffic offloading, renewable energy integration, clustering, the hybridization of solutions, and sophisticated algorithms such as reinforcement learning and genetic algorithms.

While these solutions offer promising avenues for energy conservation, they also underscore the inherent complexities and challenges in attaining optimal energy efficiency. The trade-offs between energy savings, Quality of Service (QoS), computational complexity, and infrastructure requirements necessitate careful consideration and adaptation to specific network scenarios. For instance, reinforcement learning-based approaches, while adaptive, require a learning phase that could impact immediate deployment. Similarly, certain algorithms, though effective, may exhibit high computational complexity, potentially hindering scalability in large-scale networks.

The future trajectory of energy-efficient 5G and beyond 5G networks hinges on the continuous exploration and refinement of these solutions. Addressing limitations such as the learning phase in reinforcement learning, computational complexity of certain algorithms, and need for specialized infrastructure will be pivotal. The integration of online algorithms, which can adapt to real-time network dynamics, alongside clustering techniques, renewable energy sources, traffic offloading to lower consumption networks, and the hybridization of all these, presents a promising direction for further research. The dynamic clustering of base stations based on real-time traffic patterns and cooperation factors can enable more granular control over energy-saving mechanisms, while the incorporation of renewable energy sources can contribute to a greener and more sustainable network ecosystem.

The successful realization of energy-efficient strategies will necessitate a holistic approach that considers the dynamic nature of networks, users’ diverse and evolving demands, and the continuous advancements in technology. The pursuit of sustainable and high-performing networks will remain a central focus as we navigate the complexities of the ever-evolving telecommunications landscape. The convergence of cutting-edge research, real-world data-driven insights, and collaborative efforts across industry and academia will be instrumental in shaping the future of energy-efficient 5G and beyond 5G networks.

## Figures and Tables

**Figure 1 sensors-24-07402-f001:**
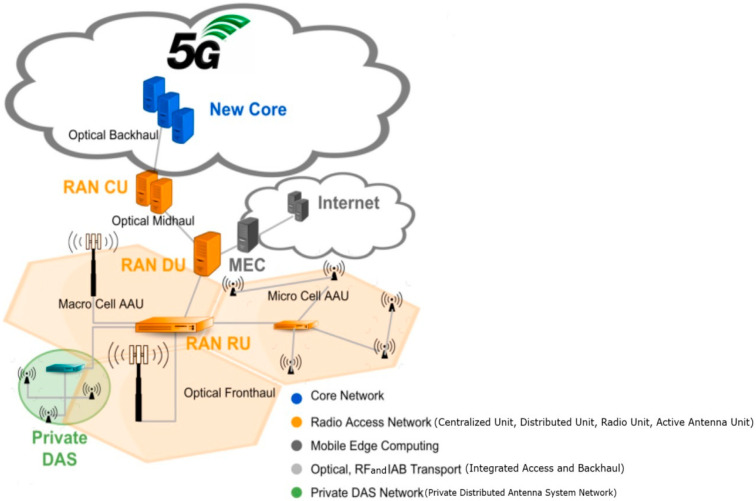
5G network.

**Figure 2 sensors-24-07402-f002:**
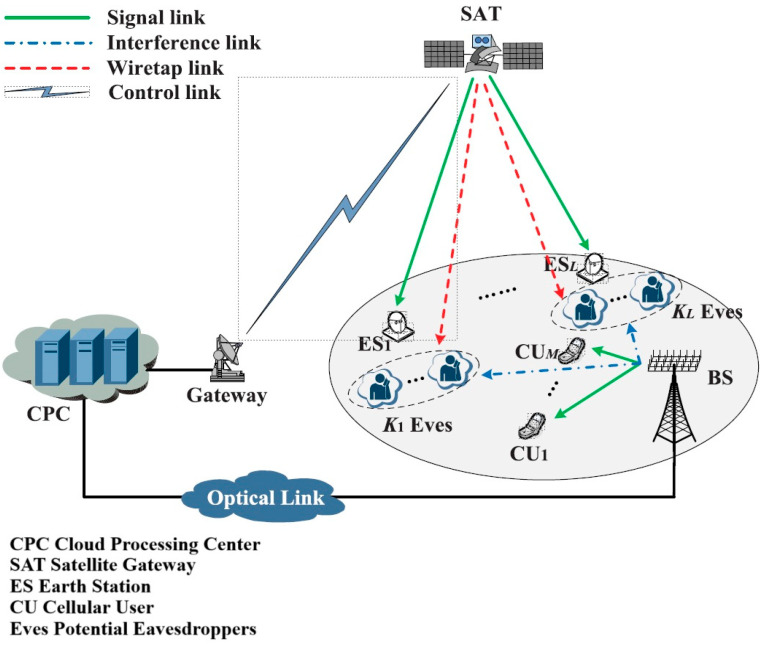
System model of the considered STIN.

**Figure 3 sensors-24-07402-f003:**
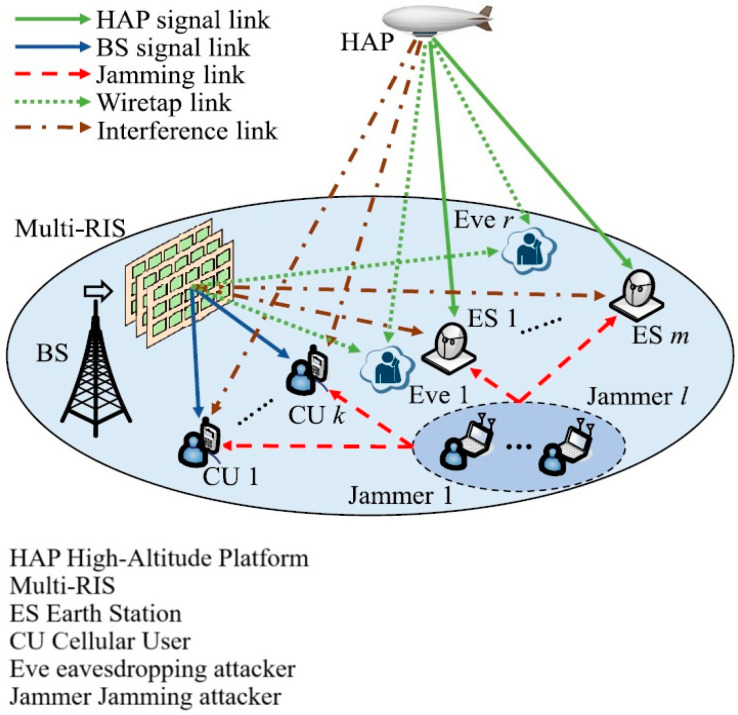
System model of ITAN with multi-layer RIS.

**Figure 4 sensors-24-07402-f004:**
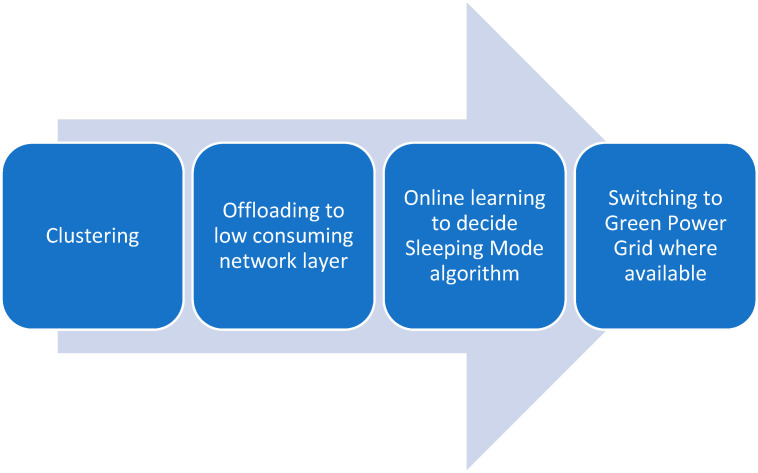
Proposed theoretical solution.

**Table 1 sensors-24-07402-t001:** Features of energy efficiency studies of 5G networks.

Feature	Study 1 [9]	Study 2 [10]	Study 3 [11]	Study 4 [12]	Study 5 [14]	Study 6 [15]	Study 7 [13]	Study 8 [16]	Study 9 [17]	Study 10 [18]	Study 11 [19]	Study 12 [20]	Study 13 [21]	Study 14 [22]
Mobility Prediction	No	Yes, semi-Markov model and landmarks	Yes, considers user positions and velocities	No	No	No	No	No	No	No	No	No	No	No
Interference		No	No	No	No	No	No	No	No	No	No	No	Yes	No
Traffic Offloading	Yes, considers dynamic offloading	No	No	Yes, considers offloading impact	No	Yes, considers load balancing.	No	No	Yes, load-based switching	No	Yes	Yes	Yes	Yes
Sleep Modes	Yes, Advanced Sleep Modes	Yes, on/off sleep modes	Yes, simple on/off switching	Yes, multi-level SMs	Yes, BS switching on/off	Yes, sleep mode for lightly loaded BSs	Yes, four sleep modes	Yes, binary on/off states for BSs	No	Yes, Advanced Sleep Modes	No	Yes	Yes	Yes, based on their traffic load and type (macro cell or small cell)
Renewable Energy	Integrates solar power	No	No	No	No	Integrates solar power	No	No	No.	No	No	No	No	No
Clustering	No	No	No	No	No	No	No	No	No	No	Yes	Yes	No	No
Algorithm	Stochastic	Genetic, Heuristic	Markov Decision Process (MDP)	Reinforcement Learning	Integer Linear Program (ILP)	Heuristic	SARSA	ESGA-5G	Heuristic	DCASM	Heuristic	Genetic	Heuristic	START5
Evaluation Metrics	Energy gain, consumption, blocking probability, reactivation delay, cost	Energy Reduction Gain	Energy consumption, QoS metrics	Energy consumption, data loss, delay.	Energy consumption, throughput, energy efficiency, no of switched-off BSs	Energy consumption gain, load factor, energy saving index, energy efficiency, spectral efficiency, radio efficiency	Energy savings, latency, QoS for different use cases	Energy savings	Energy savings	Energy savings, average BS reactivation delay	Energy efficiency, throughput, total power consumption, no of turned-off BS	No. of active base stations, energy efficiency, algorithm complexity, probability of user connection dropped	Energy efficiency, throughput, power consumption, inimum Individual EE of Active RUs	Power saving gain, convergence, network densification, load factors
EE improvements	Up to 65%	Up to 68%	Between 20 to 60%	Not evaluated	Between 22 to 33%	Up to 69%	Up to 80%	Between 28 to 54%	Up to 32%	Up to 85%	Not evaluated	Between 2–2.7%	Between 30 to 60%	Between 25 to 65%

**Table 2 sensors-24-07402-t002:** Features of studies of beyond 5G new networks.

Features	Study 1 [23]	Study 2 [24]	Study 3 [25]
Metrics	Maximize SEE	Maximize SEE	Improve secrecy
Optimization variable	Hybrid BF	Hybrid BF	BF, Alternating Optimization
Takes into consideration imperfect CSI	Yes, for wiretap channels	Yes, for jamming and wiretap channels	Yes, for direct and cascaded channels of the legitimate and eavesdropping links
Other constraints	SINR requirements, analog precoder power, transmit power	Target rate requirements, wiretap rate requirements, transmit power constraints, RIS unit-modula constraint	Worst case QoS constraints and the UMC of the phase shifter
Use of SCA	Yes	Yes	Yes
Charnes–Cooper approach	Yes	No	No
Use of interference	No	Yes	No
EE improvements	NA	0.0814 and 0.1190 bps/Hz/Joule	NA

**Table 3 sensors-24-07402-t003:** Advantages and limitations of energy efficiency studies of 5G networks.

Study	Strength	Limitation	Real Network Deployment Considerations
Study 1 [9]	Comprehensive, multiple techniques, green sources	Assumes renewable power source availability, complex infrastructure	Assumes ideal solar conditions, may require complex energy management systems
Study 2 [10]	Proactive, considers mobility, minimizes wastage from idle cells	Relies on accurate mobility prediction	Not explicitly addressed, but mentions centralized control and potential for scalability
Study 3 [11]	Dynamic, adaptive to real-time user behavior	High computational complexity	Not explicitly addressed, but computational complexity could be a challenge
Study 4 [12]	Distributed, adaptable to traffic/interference	Learning phase required, potential for suboptimal policies	Requires real-time monitoring and adaptation to network conditions
Study 5 [14]	Provides optimal (ILP) and heuristic (GA) solutions, considers BS switching and user QoS	Complexity of ILP, GA needs parameter tuning	Implementation complexity for joint optimization and BS switching
Study 6 [15]	Considers dynamic point selection CoMP, load balancing, cell zooming	Heuristic approach may not guarantee global optimality	Requires coordination between BSs and central controller
Study 7 [13]	Adaptive to traffic patterns, balances energy saving and QoS	Requires training, may not generalize to all scenarios	Real-time traffic estimation and BS control needed
Study 8 [16]	Simple, efficient, good energy-saving performance	May not be optimal for complex network topologies	Practicality of frequent BS switching, impact on QoS
Study 9 [17]	Adaptive to traffic load, balances energy efficiency & data rate	Requires accurate traffic prediction, potential for handover overhead	Requires coordination between fronthaul and backhaul, potential for increased complexity
Study 10 [18]	Simple, adaptable to traffic patterns, guarantees delay constraints	Limited to low traffic periods, assumes Poisson arrivals	Requires real-time traffic estimation and prediction
Study 11 [19]	Suitable for dense small cell networks using clustering approach reducing complexity, adaptability to non-uniform UE distribution	Static network assumption -> may need to re-run to obtain the optimal, focus on downlink throughput, does not take into consideration mobility prediction	Not explicitly considered
Study 12 [20]	Reduced complexity, centralized solution, GA, dynamic clustering, reduced complexity	Assumption of full coverage, limited consideration on mobility; centralized solution (SPOF)	Use of traffic patterns from real networks but do not consider impact of real network deployments
Study 13 [21]	Energy efficient, stable performance with variety of user density and traffic load, interference management, QoS awareness, linear complexity of the algorithm	Does not take into account latency, reliability, or mobility	Does not explicitly consider aspects such as the computational complexity of the algorithm in large-scale networks or the potential impact of channel variations on the algorithm’s performance
Study 14 [22]	Adaptability to network conditions, consideration of Node Type, minimization of control plane signaling, enhanced network robustness, applicability in real world scenarios	Implementation complexity, simplified power model, assumption of full coverage, limited consideration of mobility, centralized approach	Scalability is thought to be based on minimizing signaling; simulation based on different traffic profiles and using real network topologies

**Table 4 sensors-24-07402-t004:** Studies’ comparisons of beyond 5G new networks.

Comparison	Study 1 [23]	Study 2 [24]	Study 3 [25]
Advantages	Improved SEE, robust to imperfect AoD, CSI	Improved SEE, enhanced security, robust to CSI, flexible and adaptable	Improved secrecy, robust to CSI
Disadvantages	Limited to STIN scenarios	Increased complexity due to RIS, computation intensive for large networks	High computational

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
