# Peer review of "Energy Efficiency for 5G and Beyond 5G: Potential, Limitations, and Future Directions"

_sensors, 2024, doi:10.3390/s24227402_

Round 1

Reviewer 1 Report

Comments and Suggestions for Authors

I suggest the authors to continue to deal with this topic and expand their research. I suggest the authors to name the table 1, indicate what it represents so that the readers would understand more quickly what the authors want to show (Detailed markings on the table itself)

-- 
Additional comment

This  paper presents an exhaustive review of power-saving research conducted for 5G and beyond 5G  networks in recent years, elucidating the advantages, disadvantages, and key characteristics of each technique. Reinforcement Learning, heuristic algorithms, genetic algorithms, Markov decision processes, and the hybridization of various standard algorithms inherent to 5G and 5G NR represent a subset of the available solutions that shall undergo scrutiny. This paper identifies key limitations, namely computational expense, deployment complexity, or scalability constrai.

 I consider relevant an exhaustive review of energy saving research conducted for 5G and beyond 5G networks in recent years, as well as identification of key limitations, computational costs, implementation complexity.

 Just a comprehensive review of energy saving research for 5G and beyond 5G adds even more information to the subject area.

 I suggest the authors to name the table 1, indicate what it represents so that the readers would understand more quickly what the authors want to show (Detailed markings on the table itself)

Conclusions are aligned with a thorough review of research, as well as with arguments.
 Make the table x rib more correct.

 The literature used in the paper is relevant and innovative.

Reviewer 2 Report

Comments and Suggestions for Authors

Please look at the attached file to see the reviewer's comment 

Reviewer 3 Report

Comments and Suggestions for Authors

Point wise review for the manuscript byere is a more critical

The big question that this research tries to respond to is:

Critique: This manuscript addresses energy efficiency in 5G networks and the question asked here is relevant. It could do better in outlining the main research question at the beginning though. The other focus is energy efficiency, but a bit more detail about what in particular makes the proposals so different from those that already exist would have been great for this paper. Moreover, greater emphasis on the potential shortcomings of current energy-saving methods and how research could solve these might enhance the impact.

How original and important is the question to theory, methods or substantive results in our field?

Analysis: The critical evaluation of the considered paper is that in addition to combining various energy-saving techniques for 5G networks, this knowledge work has also not brought a high degree of originality to its domain. There are several proposed methods, which have been studied in prior work such as advanced sleep modes and clustering. And backing these to less delved areas would heighten the prospect of relevance in this manuscript, say — challenges regarding practical deployment or real-world constraints affecting their implementation. This is both a glaringly obvious point around what makes this research unique and some low hanging fruit to make it more valuable.

Relevance of the Topic to Researchers in this Subject Area:

Review: The contribution is a bit more evolutionary than revolutionary. This paper is a well-structured synthesis, but does not offer a new big idea or game-changing improvement over current methods. For example, reinforcement learning and heuristic algorithms have already been well researched in related works [rawal2012analysis] but the paper does not provide significant evidence to prove that their use of these techniques are better than the existing ones. The contribution could be more convincing if there were direct comparisons to state-of-the-art methods or rigorous evaluations on performance.

Improvements in Methodology:

Critique: The method section is not lengthy or rigorous enough to make this paper truly stand out. Although simulation based approach is a legitimate tool, it does not provide enough information about the range of parameters for which these methods can be scaled. It would be worthwhile to evaluate it more strenuously (e.g., through extensive real-world testing or a richly-varied simulation environment containing different network scenarios: traffic conditions, mobility patterns and environmental factors). In addition, the paper could detail hybrid methods to complement reinforcement learning with other optimization methods for a more complete solution.

Consistency of Conclusions:

Critical Review: The data presented generally supports the conclusions, but these are not placed in wider context for a critical review of their implications. However, the paper would benefit from engaging in a more critical discussion of its results and what they suggest regarding how well these techniques work or do not on real-world asset data. For example, the potential energy efficiency versus QoS trade-offs are mentioned but not explored well enough Furthermore, the paper does not fully explore practical questions about scalability concerns in actual network deployments.

Relevance of References

Critical Review: References are appropriate for the most part, but many of them refer to simulation-based studies and theoretical work. There should be more practical examples from the real world and data or results of empirical research on energy-efficient 5G. It should also refer more to newer studies or ones in the realm of beyond 5G and how that will support its statements on future research. An attached critique of the works refrenced, addressing their downfalls and how this paper makes up for them would also take this contribution to a more sophisticated scholarly level.

Tables and Figures: When should they be used?

Critical review : clear but quite basic tables and figures This could be widened to provide more comparison data, perhaps exhibiting benchmarks not solely on power savings however additionally latency, cost and effectiveness intricacy. What is more, the methodology could be accompanied with visualisation of the algorithms (e.g. by flowcharts or block diagrams), so as to assist in comprehension for readers who do not know their functionality beforehand. Detailed results, comparing the performance of proposed solutions with existing ones would greatly improve this paper.

Overall Critical Feedback:

Though the manuscript provides a comprehensive survey of potential energy-saving solutions for 5G and beyond 5G networks, it lacks more convincing description on contribution. It is the form of solution, and a simple theoretical analysis without complete real-world deployment scenarios or rigorous performance evaluation so that it cannot achieve realization. The paper would also improve from a more critical exploration of the shortcomings in current techniques and how these proposed methods can directly address such gaps. The paper should have been more rigorous in methodology and also provided additional comparative data to be of value.

Comments on the Quality of English Language

improved

Round 2

Reviewer 2 Report

Comments and Suggestions for Authors

The authors have addressed my concerns, no further comments.

Author Response

Thank you for great suggestions!

Reviewer 3 Report

Comments and Suggestions for Authors

The paper identifies four high-level technical challenges pertinent to energy efficiency of 5G and beyond.

Concerns of Complexity and Scalability: A number of reinforcement learning-based and heuristic algorithms, are reported to have high computation complexity, which makes the scalability of those approaches difficult for large-scale real-world applications​

Reliance on Correct Predictions: Approaches based on mobility and traffic prediction (e.g., MDPs and reinforcement learning) require accurate predictions about user movement and load. These inaccuracies can result in poor energy management and therefore lower efficiency levels.

Practical Deployment Constraints: Certain methods that include for example renewable energy sources along with a complex set of infrastructure (like clustering algorithms) take an ideal-condition assumption that is not applicable in an actual network scenario. Which limits their applicability and real-world deployments​ effectiveness.

Tackling Energy Efficiency and QoS: energy saving gains are hard to keep in pace with the QoS degradation and Similarly QoS improvement does not provides energy efficiency guarantee which is a well known issue. There are no purely happy answers that come with out a trade-off, and aggressive energy-saving measures​ can induce prolonged latency, low facts quotes, and a non permanent degradation of provider in periods of excessive demand over time​.

please review the following articles that show how 5G is used in this domain:

-Efficient Blockchain-Based Pseudonym Authentication Scheme Supporting Revocation for 5G-Assisted Vehicular Fog Computing

-FCA-VBN: Fog computing-based authentication scheme for 5G-assisted vehicular blockchain network

Comments on the Quality of English Language

need improved
